# FD-GAN: Pose-guided Feature Distilling GAN for Robust Person Re-identification

Yixiao Ge[1]*     Zhuowan Li[2,3]*†     Haiyu Zhao[2]     Guojun Yin[2,4]†
Shuai Yi[2]     Xiaogang Wang[1]     Hongsheng Li[1] ‡

[1]CUHK-SenseTime Joint Laboratory, The Chinese University of Hong Kong
[2]SenseTime Research     [3]Johns Hopkins University
[4]University of Science and Technology of China
{yxge@link, hsli@ee, xgwang@ee}.cuhk.edu.hk
{zhaohaiyu, yishuai}@sensetime.com
zli110@jhu.edu     gjyin@mail.ustc.edu.cn

## Abstract

Person re-identification (reID) is an important task that requires to retrieve a person's images from an image dataset, given one image of the person of interest. For learning robust person features, the pose variation of person images is one of the key challenges. Existing works targeting the problem either perform human alignment, or learn human-region-based representations. Extra pose information and computational cost is generally required for inference. To solve this issue, a Feature Distilling Generative Adversarial Network (FD-GAN) is proposed for learning identity-related and pose-unrelated representations. It is a novel framework based on a Siamese structure with multiple novel discriminators on human poses and identities. In addition to the discriminators, a novel same-pose loss is also integrated, which requires appearance of a same person's generated images to be similar. After learning pose-unrelated person features with pose guidance, no auxiliary pose information and additional computational cost is required during testing. Our proposed FD-GAN achieves state-of-the-art performance on three person reID datasets, which demonstrates that the effectiveness and robust feature distilling capability of the proposed FD-GAN. ‡‡

## 1 Introduction

Person re-identification (reID) is a challenging task, with the purpose of matching pedestrian images with the same identity across multiple cameras. With the wide usage of deep learning methods, reID performances by different algorithms increase rapidly. There are various attempts on learning representations with deep neural networks, however, posture variations, blur and occlusion still pose great challenges for learning discriminative features. Two types of methods were used for addressing the issues, aligning pedestrian images [1] or integrating human pose information by learning body-region features [2]. However, these works also require auxiliary pose information in the inference stage, which limits the generalization of the algorithms to new images without pose information. Meanwhile, the computational cost increases due to more complicated inference of pose estimation.

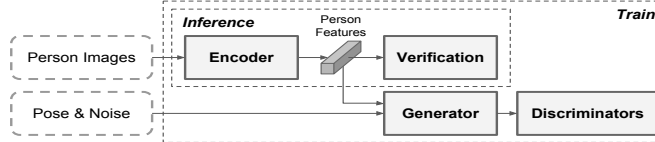

Figure 1: The image encoder in the FD-GAN is trained to learn robust identity-related and pose-unrelated representations with assistance of pose-guided image generator and discriminators. During inference, it does not need pose information and additional computational cost.

Generative adversarial network (GAN) is gaining increasing attention for image generation. Recently, some works exploited GANs' potential on aiding current person reID algorithms. Zheng *et al*. [3] proposed a semi-supervised structure which learns generative images with label smoothing regularization for outliers (LSRO) regularization. PTGAN [4] was proposed to bridge the domain gap between separate datasets. In addition to image synthesis, GAN can be used for representation learning as well. In this work, we propose a novel identity-related representation learning framework for robust person re-identification.

The proposed Feature Distilling Generative Adversarial Network (FD-GAN) maintains identity feature consistency under pose variation without increasing the complexity of inference (illustrated in Figure 1). It adopts a Siamese structure for feature learning. Each of the branch consists of an image encoder and an image generator. The image encoder embeds person visual features given the input images. The image generator generates new person images conditioned on the pose information and the input person features by the encoder. Multiple discriminators are integrated in the framework to distinguish inter-branch and intra-branch relations between generated images by the two branches.

The proposed identity discriminator, the pose discriminator, and the verification classifier together with a reconstruction loss and a novel same-pose loss jointly regularizes the feature learning process for achieving robust person reID. With the adversarial losses, identity-irrelevant information, such as pose and background appearance, in the input image is mitigated from the visual features by the image encoder. More importantly, during inference, additional pose information is no longer needed and saves additional computational cost. Our method outperforms previous works in three widely-used reID datasets, *i.e.* Market-1501 [5], CUHK03 [6] and DukeMTMC-reID [7] datasets.

Overall, this paper has the following contributions. 1) We propose a novel framework, FD-GAN, to learn identity-related and pose-unrelated representations for person re-identification with pose-variation. Unlike existing alignment or region-based learning methods, our framework does not require extra auxiliary pose information or increase the computational complexity during inference. 2) Although person image generation is an auxiliary task for our framework, the generated person images by our proposed method show better quality than existing specific person-generation methods. 3) The proposed FD-GAN achieves state-of-the-art re-identification performance on Market-1501 [5], CUHK03 [6], and DukeMTMC-reID [7] datasets.

## 2   Related Work

**Generative Adversarial Network (GAN).** Goodfellow *et al*. [8] first introduced the adversarial process to learn generative models. The GAN is generally composed of a generator and a discriminator, where the discriminator attempts to distinguish the generated images from real distribution and the generator learns to fool the discriminator. A set of constraints are proposed in previous works [9, 10, 11, 12] to improve the training process of GANs, e.g., interpretable representations are learned by using additional latent code in [12]. GAN-based algorithms shows excellent performance in image generation [13, 14, 15, 16, 17]. In terms of person image generation, PG$^2$ was proposed to synthesize person images in arbitrary poses in [18]. Siarohin *et al*. [19] designed a single-stage approach with deformable skip connections in the generator for better deformable human generation. Zanfir *et al*. [20] transferred the appearance from the source image onto the target image while preserving the target shape and clothing segmentation layout. In contrast, our method aims at learning person features for person reID with assistance of GANs. Apart from person image synthesis, pose-disentangled representations were learned for face recognition by DR-GAN [21], which has key differences with our method. Our experimental results show that our proposed FD-GAN performs

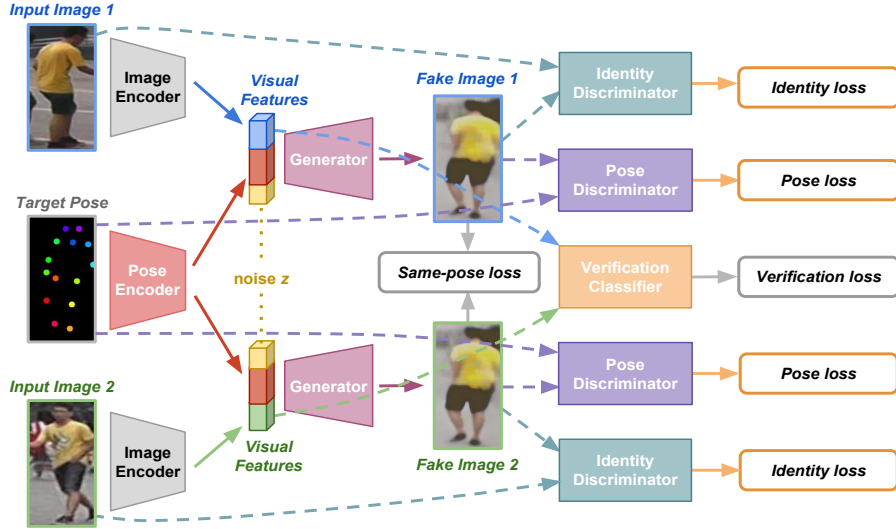

Figure 2: The Siamese structure of the proposed FD-GAN. Robust identity-related and pose-unrelated features are learned by the image encoder $E$ with a verification loss and the auxiliary task of generating fake images to fool identity and pose discriminators. A novel same-pose loss term is introduced to further encourage learning identity-related and pose-unrelated visual features.

better than DR-GAN on person reID. Our main goal is to decompose the pose information from the image features via adversarial training for learning identity-related and pose-unrelated representation.

**Person Re-identification (ReID).** Person reID [22, 23, 24, 25, 26, 27, 28, 29] is a challenging task due to various human poses, domain differences, occlusions, *etc*. Two main types of methods were adopted in previous works, *i.e.* learning discriminative person representations [30, 31, 32] and metric learning [25, 33, 34, 35]. PAN [1] aligns pedestrians and learn person features simultaneously without any extra annotation. Zhao *et al*. [2] proposed SpindleNet for learning person features of different body regions with additional human pose information. Most recent methods [1, 25, 33, 34, 35, 36, 37, 38, 39] designed more complicated frameworks to learn more robust representations with increasing computational cost or requiring extra information during inference.

Inspired by the excellent performances of GAN-based structures for image generation, there were previous works [3, 21, 4] begining to design GAN-based algorithms to improve verification performance of person reID. Zheng *et al*. [3] introduced a semi-supervised pipeline for jointly training generated images and real images from training dataset by the proposed LSRO method for regularizing unlabelled data. PTGAN [4] was proposed to bridge the domain gap between separate person reID datasets. Due to the challenges from pose diversity for person reID datasets, we propose an novel GAN-based framework for distilling identity-related features.

## 3    Feature Distilling Generative Adversarial Network

Our proposed Feature Distilling Generative Adversarial Network (FD-GAN) aims at learning identity-related and pose-unrelated person representations, in order to handle large pose variations across images in person reID.

The overall framework of our proposed method is shown in Fig. 2. The proposed FD-GAN adopts a Siamese structure, including an image encoder $E$, an image generator $G$, an identity verification classifier $V$ and two adversarial discriminators, i.e., the identity discriminator $D_{id}$ and the pose discriminator $D_{pd}$. For each branch of the network, it takes a person image and a target pose landmark map as inputs. The image encoder $E$ at each branch first transforms the input person image into feature representations. An identity verification classifier is utilized to supervise the feature learning for person reID. However, using only the verification classifier makes the encoder generally encode not only person identity information but also person pose information, which makes the learned features sensitive to person pose variation. To make the learned features robust and

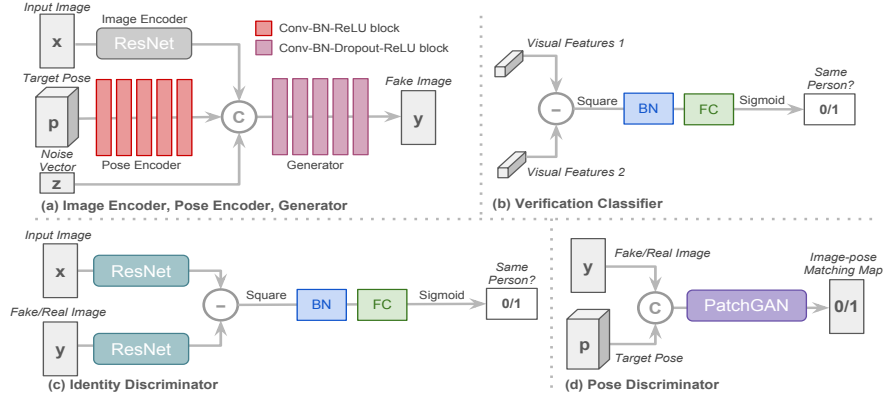

Figure 3: Network structures of (a) the generator $G$ and the image encoder $E$, (b) the verification classifier $V$, (c) the identity discriminator $D_{id}$, (d) the pose discriminator $D_{pd}$.

eliminate pose-related information, we added an image generator $G$ conditioned on the features from the encoder and a target pose map. The assumption is intuitive, if the learned person features are pose-unrelated and identity-related, then it can be used to accurately generate the same person's image but with different target poses. An identity discriminator $D_{id}$ and a pose discriminator $D_{pd}$ are integrated to regularize the image generation process. Both $D_{id}$ and $D_{pd}$ are conditional discriminators that classify whether the input image is real or fake conditioned on the input identity or pose. They are not used to classify different identities and poses. The image generator together with the image encoder are encouraged to fool the discriminators with fake generated images. Taking advantages of the Siamese structure, a novel same-pose loss minimizing the difference between the fake generated images of the two branches is also utilized, which is shown to further distill pose-unrelated information from input images. The entire framework is joint trained in an end-to-end manner. For inference, only the image encoder $E$ is used without auxiliary pose information.

## 3.1 Image encoder and image generator

The structures of the image encoder $E$ and image generator $G$ are illustrated in Figure 3(a). Given an input image $x$, the image encoder $E$ utilizes ResNet-50 as backbone network to encode the input image into a 2048-dimensional feature vector. The image generator $G$ takes the encoded person features and target pose map as inputs, and aims at generating another image of the same person specified by the target pose. The target pose map is represented by an 18-channel map, where each channel represents the location of one pose landmark's location and the one-dot landmark location is converted to a Gaussian-like heat map. It is encoded by a 5-block Convolution-BN-ReLU sub-network to obtain a 128-dimensional pose feature vector. The visual features, target pose features, and an additional 256-dimensional noise vector sampled from standard Gaussian distribution are then concatenated and input into a series of 5 convolution-BN-dropout-ReLU upsampling blocks to output the generated person images.

## 3.2 Identity verification classifier

Given the visual features of the two input images from the image encoder, the identity verification classifier $V$ determines whether the two images belong to the same person. Person identity verification is the main task for person re-identification and ensures learned features to capture identity information of person images. The structure of the classifier is shown in Figure 3(b), which takes visual features of two person images as inputs and feeds them through element-wise subtraction, element-wise square, a batch normalization layer, a fully-connected layer, and finally a sigmoid non-linearity function to output the probability that the input image pair belongs to the same person. This classifier is trained with binary cross-entropy loss. Let $x_1$, $x_2$ represent the two input person images, and $d(x_1, x_2)$ represents the output same-person confidence score by our sub-network. The identity verification classifier $V$ is trained with the following binary cross-entropy loss,

$$\mathcal{L}_v = -C \log d(x_1, x_2) - (1 - C)(1 - \log d(x_1, x_2)), \tag{1}$$

where $C$ is the ground-truth label. $C = 1$ if $x_1, x_2$ belong to the same person and $C = 0$ otherwise.

### 3.3 Image generation with identity and pose discriminators

To regularize the image encoder $E$ to learn only identity-related information, the following person image generator $G$ is trained with the identity discriminator $D_{id}$ and the pose discriminator $D_{pd}$ to generate person images with target poses. Given the input image $x_k$ ($k = 1$ or 2 for two branches) and the target pose $p$, the generated image $y_k$ is required to have the same person identity with $x_k$ but with the target pose $p$. The identity discriminator is utilized to maintain identity-related information in the encoded visual features, while the pose discriminator aims to eliminate pose-related information from the features.

**Identity discriminator** $D_{id}$ is trained to distinguish whether the generated person image and the input person image of the same branch belong to the same person. The image generator would try to fool the identity discriminator to ensure the encoded visual feature contains sufficient identity-related information. The identity discriminator sub-network has a similar network structure (see Figure 3(c)) to the identity verification classifier $V$. However, its ResNet-50 sub-network for visual feature encoding does not share weights with that of our image encoder $E$, because the identity discriminator $D_{id}$ aims at distinguishing the identity between the real/fake images, while our image encoder targets at learning pose-unrelated person features. There is domain gap between the two tasks and sharing weights hinders feature learning process of the image encoder. Such an argument is supported by our experiments. Let $y'_k$ represent the real person image having the same identity with input image $x_k$ and the target pose $p$. The adversarial loss of the identity discriminator $D_i$ can then be defined as

$$\mathcal{L}_{id} = \max_{D_{id}} \quad \sum_{k=1}^{2} \left( \mathbb{E}_{y'_k \in \mathcal{Y}}[\log D_{id}(x_k, y'_k)] + \mathbb{E}_{y_k \in \mathcal{Z}}[\log(1 - D_{id}(x_k, y_k))] \right), \tag{2}$$

where $\mathcal{Y}$ and $\mathcal{Z}$ represent the true data distribution and generated data distribution by the image generator $G$.

**Pose discriminator** $D_{pd}$ is proposed to distinguish whether the generated person image $y_k$ (for $k = 1$ or 2) matches the given target pose $p$. The sub-network structure of pose discriminator is shown in Figure 3(d). It adopts the PatchGAN [40] structure. The input image and pose map (after Gaussian-like heat-map transformation) is first concatenated along the channel dimension and then processed by 4 convolution-ReLU blocks and a sigmoid non-linearity to obtain an image-pose matching confidence map with values between 0 and 1. Each location of the confidence map represents the matching degree between the input person image and the pose landmark map. The image generator $G$ would try to fool the pose discriminator $D_{pd}$ to obtain high matching confidences with fake generated images. The adversarial loss of $D_{dp}$ is then formulated as

$$\mathcal{L}_{pd} = \max_{D_{pd}} \quad \sum_{k=1}^{2} \left( \mathbb{E}_{y'_k \in \mathcal{Y}}[\log D_{pd}([p, y'_k])] + \mathbb{E}_{y_k \in \mathcal{Z}}[\log(1 - D_{pd}([p, y_k]))] \right), \tag{3}$$

where $D_{pd}$ utilizes the concatenated person image and pose landmark map as inputs.

However, we observe that the pose discriminator $D_{pd}$ might overfit the poses, i.e., $D_{pd}$ might remember the correspondences between specific poses and person appearances, because each image's pose is generally unique. For instance, if we use a blue-top person's pose as the target pose, the generated image of a red-top person might end up having blue top. To solve this problem, we propose an online pose map augmentation scheme. During training, for each pose landmark, its 1-channel Gaussian-like heat-map is obtained with a random Gaussian bandwidth in some specific range. In this way, we can create many pose maps for the same pose and mitigate the pose overfitting problem.

**Reconstruction loss.** The responsibility of $G$ is not only confusing the discriminators, but also generating images that are similar to the ground-truth images. However, the discriminators alone cannot guarantee generating human-perceivable images. Therefore, a reconstruction loss is introduced to minimize the $L1$ differences between the generated image $y_k$ and its corresponding real image $y'_k$, which is shown to be helpful for more stable convergence of training the generator.

$$\mathcal{L}_r = \sum_{k=1}^{2} \frac{1}{mn} \|y_k - y'_k\|_1, \tag{4}$$

where $mn$ is the number of pixels in the real/fake images. When there is no corresponding ground-truth image $y'_k$ for an input image $x_k$ and a target pose $p$, this loss is not utilized.

**Same-pose loss.** The purpose of the image generator $G$ is to help the image encoder distill only pose-unrelated information. We input the same person's two different images and the same target pose to both branches of our Siamese network, if the conditioning visual features in the two branches are truly only identity-related, then the two generated images should be similar in appearance. Therefore, we propose a same-pose loss to minimize the differences between the two generated images of the same person and with the target pose,

$$\mathcal{L}_{sp} = \frac{1}{mn}\|y_1 - y_2\|_1, \tag{5}$$

which encourages the learned visual features from $E$ of the two input images to only be identity-related while ignoring other factors.

**The overall training objective.** The above mentioned classifier loss, discriminator losses and reconstruction losses work collaboratively for learning identity-related and pose-unrelated representations. The overall loss function could be defined by

$$\mathcal{L} = \mathcal{L}_v + \lambda_{id}\mathcal{L}_{id} + \lambda_{pd}\mathcal{L}_{pd} + \lambda_r\mathcal{L}_r + \lambda_{sp}\mathcal{L}_{sp}, \tag{6}$$

where $\lambda_{id}$, $\lambda_{pd}$, $\lambda_r$, $\lambda_{sp}$ are the weighting factors for the auxiliary image generation task.

### 3.4 Training scheme

There are three stages for training our proposed framework. In the first stage, our Siamese baseline model, which includes only the image encoder $E$ and identity verification classifier $V$, is pretrained on a person reID dataset with only identity cross-entropy loss $\mathcal{L}_v$ in Eq. (1). The pre-trained network weights are then used to initialize $E$, $V$, and identity discriminator $D_{id}$ in stage-II. In the second stage, parameters of $E$ and $V$ are fixed. We then train $G$, identity discriminator $D_{id}$, and pose discriminator $D_{pd}$ with the overall objective $\mathcal{L}$ in Eq. (6). Finally, the whole network is finetuned jointly in an end-to-end manner. For each training mini-batch, it contains 128 person image pairs, with 32 of them belonging to same persons (positive pairs) and 96 of them belonging to different persons (negative pairs). All images are resized to $256 \times 128$. The Gaussian bandwidth for obtaining pose landmark heat-map is uniformly sampled in $[4, 6]$.

In training stages II and III, the discriminators and other parts of the network are alternatively optimized. When jointly optimizing the generator $G$, the image encoder $E$ and the verification classifier $V$, the overall objective Eq. (6) is used. When optimizing the discriminators $D_{id}$ and $D_{pd}$, only adversarial losses $\mathcal{L}_{id}$ and $\mathcal{L}_{pd}$ are adopted.

**Stage I: ReID baseline pretraining.** Our Siamese baseline only includes the image encoder $E$ and identity verification classifier $V$. The ResNet-50 sub-network is first initialized with ImageNet-pretrained weights [41]. The network is optimized by Stochastic Gradient Descent (SGD) with momentum 0.9. The initial learning rates are set to 0.01 for $E$ and 0.1 for $V$, and they are decreased to 0.1 of their previous values every 40 epochs. The stage-I training process iterates for 80 epochs.

**Stage II: FD-GAN pretraining.** With $E$ and $V$ fixed, We integrate $G$, $D_{id}$, and $D_{pd}$ into the framework in stage-II. Adam optimizer is adopted for optimizing $G$ and SGD for $D_{id}$ and $D_{pd}$. The initial learning rates for $G$, $D_{id}$, $D_{pd}$ are set as $10^{-3}$, $10^{-4}$, $10^{-2}$, respectively. Learning rates maintain the same for the first 50 epochs, and then gradually decrease to 0 in the following 50 epochs. The loss weights are set as $\lambda_{id} = 0.1, \lambda_{pd} = 0.1, \lambda_r = 10, \lambda_{sp} = 1$. We took the label smoothness scheme [42] for better balancing between the generator and the discriminator.

**Stage III: Global finetuning.** For finetuning the whole framework end-to-end, we use Adam for optimizing $E$, $G$ and $V$, and SGD for $D_{id}$, $D_{pd}$ after loading the pre-trained weights from stage-II. Specifically, the initial learning rates are set to $10^{-6}, 10^{-6}, 10^{-5}, 10^{-4}, 10^{-4}$ for $E$, $G$, $V$, $D_{id}$, $D_{pd}$, respectively. Learning rates remain the same for the first 25 epochs, and then gradually decay to 0 in the following 25 epochs. Batch normalization layers in $E$ is fixed to achieve better performance. For loss weights, $\lambda_{id} = 0.1, \lambda_{pd} = 0.1, \lambda_r = 10, \lambda_{sp} = 1$ are set as the weights for different loss terms.
§

Table 1: Component analysis of the proposed FD-GAN on Market-1501 [5] and DukeMTMC-reID [7] datasets in terms of top-1 accuracy (%) and mAP (%)

| Networks | Components | | | | | | Market-1501[5] | | DukeMTMC-reID[7] | |
|---|---|---|---|---|---|---|---|---|---|---|
| | not share $E$ | $\mathcal{L}_{sp}$ | $\mathcal{L}_v$ | $\mathcal{L}_{pd}$ | $\mathcal{L}_{id}$ | pose map aug. | mAP | top-1 | mAP | top-1 |
| baseline (single) | n/a | n/a | n/a | n/a | n/a | n/a | 59.8 | 81.4 | 40.7 | 62.5 |
| baseline (Siamese) | n/a | n/a | √ | n/a | n/a | n/a | 72.5 | 88.2 | 61.3 | 78.2 |
| Siamese DR-GAN[21] | × | × | √ | √ | √ | √ | 73.2 | 86.7 | 60.2 | 76.9 |
| FD-GAN (share $E$) | × | √ | √ | √ | √ | √ | 73.5 | 86.8 | - | - |
| FD-GAN (no sp.) | √ | × | √ | √ | √ | √ | 75.8 | 88.9 | - | - |
| FD-GAN (no veri.) | √ | √ | × | √ | √ | √ | 75.7 | 89.5 | 62.6 | 78.8 |
| FD-GAN (no sp. & no veri.) | √ | × | × | √ | √ | √ | 74.4 | 88.7 | 62.4 | 78.6 |
| FD-GAN (no $D_{pd}$) | √ | √ | √ | × | √ | √ | 73.0 | 88.0 | - | - |
| FD-GAN (no $D_{id}$) | √ | √ | √ | √ | × | √ | 72.8 | 89.2 | - | - |
| FD-GAN (no $D_{id}$ & $D_{pd}$) | √ | √ | √ | × | × | √ | 71.6 | 84.6 | - | - |
| FD-GAN (no pose aug.) | √ | √ | √ | √ | √ | × | 77.2 | 89.5 | 63.9 | 79.5 |
| FD-GAN | √ | √ | √ | √ | √ | √ | **77.7** | **90.5** | **64.5** | **80.0** |

## 3.5 Comparison to DR-GAN [21]

There is an existing work, DR-GAN [21] based on conditional GAN [43], which tries to learn pose-invariant identity representations for face recognition. It also adopts an encoder-decoder structure with a discriminator for classifying both identity. Comparison results in Section 4.2 demonstrate the advantages of our proposed method over DR-GAN on the person reID task.

This is because there are three key differences between the proposed FD-GAN and DR-GAN, which make our algorithm superior. 1) We adopt a Siamese network structure, which enables us to use the same-pose loss to encourage encoding only learning identity-related information, while DR-GAN does not have such a loss term. 2) We do not share the weights between the ResNet-50 networks in the image encoder and in the identity discriminator. We observe that identity verification and real/fake image identity discrimination are two tasks in different domains and therefore their weights should not be shared. 3) Our Siamese structure utilizes a verification classifier instead of a cross-entropy classifier, which shows better person reID performance than a single-branch network does.

## 4 Experiments

### 4.1 Datasets and evaluation metrics

In this paper, three datasets are used for performance evaluation, including Market-1501 [5], CUHK03 [6], and DukeMTMC-reID [7]. The Market-1501 dataset [5] consists of 12,936 images of 751 identities for training and 19,281 images of 750 identities in the gallery set for testing. The CUHK03 dataset [6] contains 14,097 training images of 1,467 identities captured from two cameras. The original training and testing protocol is used. The DukeMTMC-reID dataset [7] is a subset of the pedestrian tracking dataset DukeMTMC for image-based reID. It contains 16,522 images of 702 identities for training. Mean average precision (mAP) and CMC top-1 accuracy are adopted for performance evaluation on all the three datasets.

### 4.2 Component analysis of the proposed FD-GAN

In this section, component analysis is conducted to demonstrate the effectiveness of components in the FD-GAN framework, including the Siamese structure, and the use of verification classifier and same-pose loss. We also compare with DR-GAN [21], which also proposes to learning pose disentangled features. Our Siamese baseline model is only the ResNet-50 image encoder $E$ with our identity verification classifier $V$. The analysis is conducted on Market-1501 [5] and DukeMTMC-reID [7] datasets and the results are shown in Table 1.

**Siamese structure.** We first compare the our Siamese reID baseline (denoted as baseline (Siamese)) with the single branch ResNet-50 [44] baseline trained with cross-entropy loss on person IDs (denoted as baseline (single)). The Siamese baseline outperforms single-branch baseline by 12.7% and 20.6% in terms of mAP on the two datasets.

**Proposed FD-GAN, with online pose map augmentation and adversarial discriminators.** Based on the Siamese structure, we build our proposed FD-GAN framework. We can observe that the

Table 2: Experimental comparison of the proposed approach with state-of-the-art methods on Market-1501 [5], CUHK03 [6], and DukeMTMC-reID [7] datasets. Top-1 accuracy(%) and mAP(%) are reported.

| Methods | Market-1501 [5] | | CUHK03 [6] | | DukeMTMC-reID [7] | |
|---|---|---|---|---|---|---|
| | mAP | top-1 | mAP | top-1 | mAP | top-1 |
| BoW+KISSME [5] | - | - | - | - | 12.1 | 25.1 |
| LOMO+XQDA [37] | - | - | - | - | 17.0 | 30.8 |
| OIM Loss [45] | 60.9 | 82.1 | 72.5 | 77.5 | 47.4 | 68.1 |
| MSCAN [39] | 53.1 | 76.3 | - | 74.2 | - | - |
| DCA [39] | 57.5 | 80.3 | - | 74.2 | - | - |
| SpindleNet [2] | - | 76.9 | - | 88.5 | - | - |
| k-reciprocal [46] | 63.6 | 77.1 | 67.6 | 61.6 | - | - |
| VI+LSRO [3] | 66.1 | 84.0 | 87.4 | 84.6 | - | - |
| Basel+LSRO [3] | - | - | - | - | 47.1 | 67.7 |
| OL-MANS [47] | - | 60.7 | - | 61.7 | - | - |
| PA [48] | 63.4 | 81.0 | - | 85.4 | - | - |
| SVDNet [49] | 62.1 | 82.3 | 84.8 | 81.8 | 56.8 | 76.7 |
| JLML [50] | 65.5 | 85.1 | - | 83.2 | - | - |
| Proposed FD-GAN | **77.7** | **90.5** | **91.3** | **92.6** | **64.5** | **80.0** |

proposed FD-GAN gains significant improvements from our Siamese baseline in terms of both mean AP and top-1 accuracy on both two reID datasets. There are 5.2% and 3.2% mAP improvements in terms of mAP on the two datasets. To show the effectiveness of our proposed online pose map augmentation, we test removing it when training our FD-GAN (denoted as FD-GAN w/o pose aug. in Table 1). It results in a .5% performance drop for both datasets. In order to validate the effects of the two discriminators $D_{id}$ and $D_{pd}$, we test removing them separately and together (denoted as FD-GAN w/o $D_{id}$ or $D_{pd}$. in Table 1). It results in not only obviously performance drop, but also poorer generated images.

**DR-GAN [21], verification loss, same-pose loss, and not sharing image encoder.** We also study the effectiveness of using verification loss and same-pose loss, and not sharing image encoder weights to identity discriminator. Original DR-GAN's pose discriminator classifies each face image into one of 13 poses. For fair comparison, we first test integrating DR-GAN into our Siamese baseline (denoted as Siamese DR-GAN), which could be viewed our FD-GAN without the same-pose loss and also sharing weights between $E$ and $D_{id}$. Since our network uses pose map as input condition, we use our conditional pose discriminator $D_{pd}$ to replace DR-GAN's pose discriminator. The Siamese DR-GAN even performs worse than our Siamese baseline on the DukeMTMC-reID dataset. Our proposed FD-GAN outperforms it by over 4% mAP on both datasets. We also try removing both verification classifier and same-pose loss (denoted as FD-GAN w/o sp. & veri.), removing only identity verification classifier (denoted as FD-GAN w/o veri.), removing only same-pose loss (denoted as FD-GAN w/o sp.) from our proposed FD-GAN and only sharing weights between $E$ and $D_{id}$ (denoted as FD-GAN share $E$) . Results in Table 1 show that both the verification loss and same-pose loss are indispensable to achieve superior performance on person reID. Also, not sharing weights between $E$ and $D_{id}$ results in better performance.

### 4.3 Comparison with state-of-the-arts

We compare our proposed FD-GAN with the state-of-the-art person reID methods including VI+LSRO [3], JLML [50], PA [48], *etc.* on the three datasets, Market-1501 [5] , CUHK03 [6] , and DukeMTMC-reID [7]. The results are listed in Table 2. Note that only single query results from published papers are compared in order to make a fair comparison.

By finetuning the FD-GAN based on ResNet-50 [44] baseline network structure, our proposed FD-GAN outperforms previous approaches and achieves state-of-the-art performance. We can achieve $90.5\%$ top-1 accuracy and $77.7\%$ mAP on the Market-1501 dataset [5], $92.6\%$ top-1 accuracy and $91.3\%$ mAP on CUHK03 dataset [6], and $80.0\%$ top-1 accuracy and $64.5\%$ mAP on the DukeMTMC-reID dataset [7], which demonstrates the effectiveness of the proposed feature distilling FD-GAN.

### 4.4 Person image generation and visual analysis

**Comparison of person image generation [18, 19].** Although generating person images is only an auxiliary task in our FD-GAN to learn more robust person features. We are interested in comparing

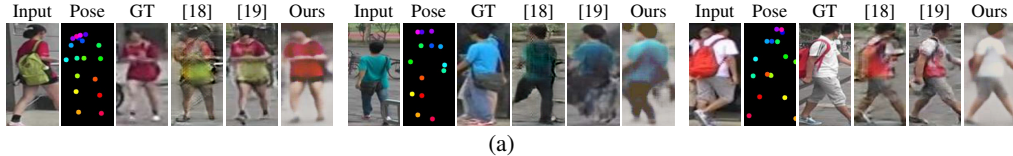

| Input | Pose | GT | [18] | [19] | Ours | Input | Pose | GT | [18] | [19] | Ours | Input | Pose | GT | [18] | [19] | Ours |

(a)

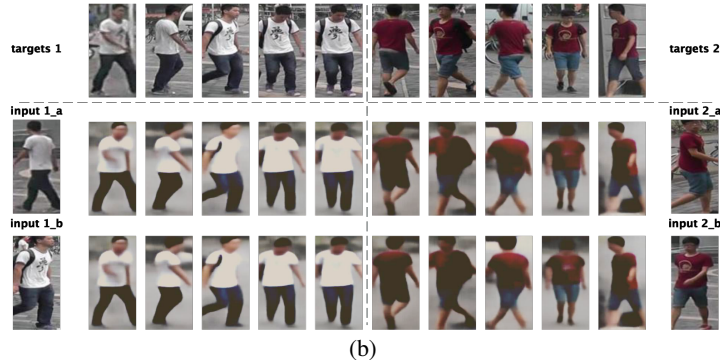

(b)

Figure 4: (a) Generated person images by our proposed method and [18, 19] on test images of Market-1501 dataset [5]. (b) Two examples of the generated images from the Market-1501 dataset [5]. (First row) the ground-truth images of target poses. (Second-third rows) input images and the generated images with different target poses on training images of Market-1501 dataset [5].

the generated images with images by other specifically designed person generation methods [18, 19]. Figure 4(a) shows the generated person images by state-of-the-art person generation methods [18, 19] and our FD-GAN. One can clearly see that our proposed method better understand the concept of "backpack" and could generate correct upper and lower body clothes. We argue that the key is using person identity supervisions to make the encoder learn better identity-related features. Our Siamese structure and the same-pose loss also contribute to achieving consistent generation results.

**Visualization for learned features.** The proposed FD-GAN framework not only improves the discriminative capability of visual features but could also be used as a visualization tool for manually examining learned feature representations. The quality of learned person features have direct impact on the generated person images. We can therefore tell what aspects of person appearances are captured by the features. For instance, for "input 1_b" in Figure 4(b), its generated frontal images do not show colored pattern on the upper body but only the general colors and shapes of the upper and lower bodies, which might demonstrate that the learned image encoder focus on embedding the overall appearances of persons but fail to capture the distinguishable details in appearance.

## 5  Conclusion

In this paper, we proposed the novel FD-GAN for learning identity-related and pose-unrelated person representations with human pose guidance. Novel Siamese network structure as well as novel losses ensure the framework learns more pose-invariant features for robust person reID. Our proposed framework achieves state-of-the-art performance on person reID without using additional computational cost or extra pose information during inference. The generated person images also show higher quality than existing specific person-generation methods.

**Acknowledgements**

This work is supported by SenseTime Group Limited, the General Research Fund sponsored by the Research Grants Council of Hong Kong (Nos. CUHK14213616, CUHK14206114, CUHK14205615, CUHK14203015, CUHK14239816, CUHK419412, CUHK14207814, CUHK14208417, CUHK14202217), the Hong Kong Innovation and Technology Support Program (No. ITS/121/15FX).

## Footnotes

*The first two authors contribute equally to this work.

†This work was done when they were interns at SenseTime Research.

‡Hongsheng Li is the corresponding author.

‡‡The code is now available. https://github.com/yxgeee/FD-GAN

§We tune hyperparameters on the validation set of Market-1501 [5], and directly use the same hyperparameters for DukeMTMC-reID [7] and CUHK03 [6] datasets.

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
