[Reviews · NeurIPS 2018]

Reviewer 1



This paper describes a GAN approach to addressing a common and important problem in person re-identification: inter- and intra-view pose variation. The technique, in extreme synthesis, uses a generative model to implicitly marginalize away pose- and background-dependent information in the feature representaiton to distill a representation that is invariant to both, but still discriminative for person identities. Pose is represented as the spatial configuration of landmarks, and during training person images conditioned on a randomly selected pose are generated from image encodings. These generated images are fed to multiple adversarial discriminators that determine if the generated image is real/false, if the pose in a real/fake image is accurate, and if two feature embeddings correspond to the same person. Experimental results are given on multiple, important benchmark datasets and show significant improvement over the state-of-the-art. Clarity, quality, and reproducibility: The clarity of exposition is quite good. The authors do a good job of describing a complicated combination of modules and losses, with in my opinion the right degree of precision and level of detail. There are some minor typos and grammar errors throughout, but not to distraction. The quality of the technical exposition and experimental evaluation is similarly high, and I appreciated how each component and loss term is methodically and incrementally presented in section 3.3. I feel like the results of the paper would be challenging to reproduce, however this is mainly due to the large number of moving parts and not to lack of clarity or missing details. On a more critical note, and related to reproducibility, generative adversarial architectures with just a *single* adversarial loss are notoriously unstable during training. The proposed system has five terms in the loss, and three phases of (what appears to be) carefully crafted training schedule. Was this training pipeline, with its myriad hyperparameters, derived on a validation re-identification dataset? More insight into the behavior of the system at training time would be a useful contribution. Also, some ablation analysis showing the incremental contributions each loss would lend credibility to the need for each of the adversarial and identity modules and losses. Novelty: This work uses the idea of generative adversarial models and applies it in a novel and clever way to learn identity discriminative feature representations that are robust to pose and background variations -- probably the two most significant challenges in person re-identification. This type of adversarial invariance learning has not been applied in person re-identification, and I am unaware of similar approaches in other applications (other than DR-GAN for pose-invariant face recognition, included in the comparative evaluation). The authors do an excellent job of positioning their work in the context of the state-of-the-art and analyzing the advantages of their approach. Significance and fit with NIPS: The experimental results on multiple, modern, and large scale re-identification benchmarks show a significant improvement over the state-of-the-art. Though the proposed system is based on multiple known techniques, the combination is novel and well thought-out. This paper is fundamentally an architecture engineering paper, but an extremely well-executed one. It is likely to have significant impact in the person re-identification community, and could potentially contribute insights to adversarial invariant feature learning in the broader NIPS world. Some specific comments and questions: 1. Figures 2 and 3 are gorgeous and do a good job of explaining the complexities of the components. However their size has been reduced to the limits of printed legibility (likely due to space). Line 19: widely --> wide Line 103: regularizes --> regularize Line 113: same person's another image --> another image of the same person Line 210: integratie --> integrate POST REBUTTAL: The rebuttal addressed my main concerns (and I was already quite positive). The clarifications about training and crossvalidation, plus the additional ablation table, make the paper even stronger.

Reviewer 2



Summary: This paper proposes a novel framework, FD-GAN, to learn identity-related and pose-unrelated representations for person re-identification with pose variation that does not require extra auxiliary pose information or increase the computational complexity during inference. FD-GAN uses a Siamese structure, including an image encoder E, an image generator G, an identity verification classifier V and two adversarial discriminators. The image encoder E at each branch first transforms the input person image into feature representations. An identity verification classifier is utilized to supervise the feature learning. An image generator G conditioned on the features from the encoder and a target pose map is added to make the learned features robust without pose-related information. An identity discriminator and a pose discriminator are integrated to regularizes the image generation process. The image generator together with the image encoder are encouraged to fool the discriminators with fake generated images. Taking advantages of the Siamese structure, a novel same-pose loss minimizing the difference between the fake generated images of the two branches is also utilized, which is shown to further distill pose-unrelated information from input images. Using a novel same-pose loss minimizing the difference between the fake generated images of the two branches, the entire framework is joint trained in an end-to-end manner. For inference, only the image encoder E is used without pose information. Experiments shows that different network components in FD-GAN is effective on Market-1501 and DukeMTMCreID datasets. Experiment comparing to other state-of-the-art person reID methods on Market-1501, CUHK03, and DukeMTMC-reID datasets shows that FD-GAN has the best performance. Comparison of generated images from FD-GAN to those from other methods also shows that FD-GAN has better generated images. Strengths: - This is a technical sound paper with enough experiments to justify the effectiveness of their approach. - This papers propose a novel Siamese network structure and novel losses to ensure the framework learns more pose-invariant features for robust person reID. - This paper is well written with good presentation of FD-GAN. Weaknesses: - In section 4.4, the number of cases used for visualization comparison to [18, 19] is small. It is unclear how [19] performs on the left and middle cases, and unclear how [18] performs on the right case. Rebuttal Response: After reading the rebuttal, it addresses my comments on lacking of visualization comparison to [18, 19]. I support accepting this paper, and I will not change my score.

Reviewer 3



This paper proposes a feature learning method using pose information based on GAN. The proposed method (FD-GAN) requires target pose maps and its corresponding real images only in the training stage. The base model of FD-GAN is a Siamese structure with identity verification and several loss functions. The identity discriminator loss aims to distinguish real vs. fake images of the same person. The pose discriminator aims to distinguish whether the generated image matches the given pose by patchGAN[35] structure. The reconstruction loss aims to minimize the difference between the generated image and real images if the corresponding ground image exists. The same pose loss aims to make the appearance of a same person’s generated images in a given pose to be similar. After the learning pose-unrelated person features with pose guidance, the test stage requires no auxiliary pose information. Strengths: + The proposed method is a bit incremental but seems to be novel. The DR-GAN[20] seems to be the closest work to obtaining a pose insensitive feature using distillation GAN. Compared with DR-GAN, FD-GAN uses pose maps and Siamese architecture. Recently, several works proposed GAN-based human image generation in the new poses, e.g. [18][19]. This works applied such human image generation to feature distillation for person re-identification in the first time. + The proposed method requires no pose estimation in the test stage. Some recent methods use pose estimation in an inference of person re-identification to handle pose variations. In contrast, the proposed method requires the pose estimation only when the training stage of the features. It will be practical for real applications since it does not increase computation costs. + Person re-identification results are better than state-of-the-art methods. + The example that shows the generated person images show the better quality than existing specific person-generation methods [18][19]. Weaknesses: I have some questions on the experiments. - Missing baseline: Performances when removing L_id, L_pd, L_r and performances when only removing one of the not share E, L_sp, L_v are not evaluated. - How the experimental comparison with DR-GAN was done is not clear enough since DR-GAN does not use the pose map for input data. - The proposed method does not require the pose estimation in the inference, so test image does not have pose maps. It is not clear if the visual analysis on Sec.4.4 is the result on training images or test images. - Why the generated images of Figure 4 (a) shows only one of the [18],[19] for each input image? - Why the [18][19] fails and the proposed method succeed? Several parts of this paper are hard to understand. -The names “Identity discriminator” and “pose discriminator” are confusing. When reading Fig.2, it looks like these losses are to distinguish different identities and poses. However, Eq.(2) seems to be the loss to distinguish real vs. fake images (of the same person), not to distinguish between different identities. Similarly, the “pose discriminator” is not to distinguish between different poses. - Line29: What is LOSO regularization? - I suggest replacing some arXiv papers with actual published works. Response to Rebuttal: The rebuttal addressed my concerns about the experiments. However, the additional ablation study was only on Market-1501 dataset, and DukeMTMC-reID dataset should be included in the final version. Also, the explanation of identity/pose discriminators should be carefully revised according to the response.